# Persistence of Coronavirus on Surface Materials and Its Control Measures Using Nonthermal Plasma and Other Agents

**DOI:** 10.3390/ijms241814106

**Published:** 2023-09-14

**Authors:** Sekar Ashokkumar, Nagendra Kumar Kaushik, Ihn Han, Han Sup Uhm, Jang Sick Park, Gyu Seong Cho, Young-Jei Oh, Yung Oh Shin, Eun Ha Choi

**Affiliations:** Plasma Bioscience Research Center, Kwangwoon University, Seoul 01897, Republic of Korea

**Keywords:** environment, human coronavirus, non-thermal plasma (NTP), persistent

## Abstract

Severe acute respiratory syndrome coronavirus-2 (SARS-CoV-2) has been responsible for the initiation of the global pandemic since 2020. The virus spreads through contaminated air particles, fomite, and surface-contaminated porous (i.e., paper, wood, and masks) and non-porous (i.e., plastic, stainless steel, and glass) materials. The persistence of viruses on materials depends on porosity, adsorption, evaporation, isoelectric point, and environmental conditions, such as temperature, pH, and relative humidity. Disinfection techniques are crucial for preventing viral contamination on animated and inanimate surfaces. Currently, there are few effective methodologies for preventing SARS-CoV-2 and other coronaviruses without any side effects. Before infection can occur, measures must be taken to prevent the persistence of the coronavirus on the surfaces of both porous and non-porous inanimate materials. This review focuses on coronavirus persistence in surface materials (inanimate) and control measures. Viruses are inactivated through chemical and physical methods; the chemical methods particularly include alcohol, chlorine, and peroxide, whereas temperature, pH, humidity, ultraviolet irradiation (UV), gamma radiation, X-rays, ozone, and non-thermal, plasma-generated reactive oxygen and nitrogen species (RONS) are physical methods.

## 1. Introduction

The World Health Organization (WHO) has reported that viral infections promote severe diseases in humans. Over the last 22 years, outbreaks of several viral infections have emerged, including those caused by severe acute respiratory syndrome coronavirus (SARS-CoV), H1N1 influenza (H1N1pdm09 virus), Middle East respiratory syndrome coronavirus (MERS-CoV), and SARS-CoV-2. COVID-19, which was first reported in Wuhan, Hubei region, China on 31 December 2019, and spread from animals to humans. It is also a zoonotic virus with bats as a natural stock [1], and the Bat-CoV-RaTG13 suggested that 96% of the genome sequence was identical to that in SARS-CoV-2 [1,2]. Human coronavirus (HCoV) contains four proteins, namely the nucleocapsid protein (N), which is further associated with three structurally enveloped proteins, a large transmembrane spike protein (S), membrane protein (M), an envelope protein (E), and structural and nonstructural proteins (Figure 1A).

The coronavirus proteins are involved in different activities: S protein is involved in coronavirus entry into host cells [3]; E protein is involved in morphogenesis and assembly of virions; M is embedded in the viral membrane [4]; and N proteins play multiple roles in the infection cycle and prepare to bind and package the viral ribonucleic acid (RNA into ribonucleoprotein (RNP) complexes [5]. The replication of SARS-CoV-2 within host cells involves several key steps, as shown in Figure 1B. While HCoV primarily affects the lungs, it can also impact other organs including the liver, heart, kidney, esophagus, bladder, ileum, and pancreas [6,7]. After viral infection, the endothelial layer is disrupted (e.g., thrombosis, hemorrhage, and edema), and these disrupted responses (e.g., autoimmune reactions or cytokine storms) cause various organ dysfunctions [8], as shown in Figure 1C. After HCoV infects human organs, the virus particles in airborne droplets are freely disseminated through the air via coughing, breathing, sneezing, and talking. Most infections are transmitted through close contact [9]. Viruses can spread directly through physical contact between an infected individual, indirect contact with contaminated surfaces or fomites, or directly through respiratory droplets or fine respiratory aerosol [10].

Some human respiratory coronaviruses, such as SARS-CoV-1, MERS, and SARS-CoV-2, can persist on surface materials. This persistence depends on several factors, including porosity, adsorption, evaporation, isoelectric point, hydrophobicity, relative humidity, temperature, and pH. HCoV can be inactivated by various chemical and physical treatments. Chemical compounds, such as ethanol, hydrogen peroxide, and sodium hypochlorite, have been successfully applied for the sterilization of personal protective equipment and surface-contaminated materials [11]. There are several disadvantages associated with using high concentrations of chemicals for complete viral elimination. Using chemicals for viral inactivation can potentially harm human health and the environment and, over time, might release toxic by-products. Physical inactivation mainly depends on radiation, thermal, and mechanical effects (plasma and ultraviolet (UV) radiation), and does not affect the environment or materials. One of the physical methods that resemble non-thermal plasma is the emerging disinfection technology. Plasma-fed gases, such as argon, helium, nitrogen, and air-generated reactive oxygenspecies (ROS) and reactive nitrogen species (RNS), can inactivate the virus.

In this review, we analyze the literature on the persistence of the virus on porous and nonporous material surfaces and characterize the role of surface material inactivation in viral infectivity using chemical and physical agents. The chemical methods particularly include alcohol, chlorine, and peroxide, whereas temperature, pH, humidity, ultraviolet irradiation (UV), gamma radiation, X-rays, ozone, and non-thermal plasma generated reactive oxygen and nitrogen species are physical methods.

### 1.1. Virus Transmission through Fomites

Fomite-containing viruses spread into the environment based on their fomite properties, virus characteristics, and surrounding environmental conditions, such as temperature and humidity. Coronavirus diffusion occurs through direct contact between individuals, indirect contact via fomites, and airborne transmission through droplets and aerosols [12]. Coronavirus droplets spread to the surrounding surfaces based on the droplet diameter, whereas some smaller airborne droplets (<5 mm in diameter) travel up to 7–8 m [13]. Viruses in respiratory droplets larger than 5 µm fall to the ground more quickly compared to those in smaller respiratory droplets of (<5 µm), which can spread over longer distances. In addition, SARS-CoV-2 can spread through airborne dust associated with microorganisms [14]. If a person touches virus-containing fomites and then their mouth or nose, they can potentially spread the virus to other places. The virus spread depends on material porosity, adsorption, evaporation, isoelectric point, and hydrophobicity.

#### 1.1.1. Porosity

Porous (paper, cloth, wood, and mask) and non-porous (plastic, stainless steel, and glass) are the main feature of viruses that persist on surface materials. Some studies have explained the persistence of viruses in porous and non-porous materials. Viruses exhibit greater persistence on non-porous materials compared to porous surfaces. Furthermore, viruses can survive on inanimate surfaces for 1 to 9 d at approximately 30 °C [15]. Notably, SARS-CoV-1 and SARS-CoV-2 persist on contaminated plastic, stainless steel, copper, and cardboard material surfaces for up to 72 h and on aerosols for 3 h [16]. Kasloff et al. [17] reported SARS-CoV-2 survival based on porous and non-porous materials. He pointed out that the N-100 and N-95 face masks reduced the log_10_ number 3 in SARS-CoV-2 virus titer within 4 to 7 d, although survival was 14 to 21 d on polyvinyl chloride (PVC) face masks and stainless steel, 7–14 d on gloves, and 1 d on cotton.

#### 1.1.2. Adsorption

The adsorption of coronaviruses on surface materials is crucial in stability, even as the adhesion process mainly consists of environmental conditions such as physiochemical, electrostatic charge of the virus, and fomite surface interactions. Virus adhesion to surface materials is based on the smoothness or roughness and degree of hydrophobicity of the deposition. The porous surface reduced the size of the absorbed liquid droplets over time and evaporated more easily than the non-porous surface. Droplets spread over the surface and the contact angle of adhesion approaches zero in porous materials, even though a thin film surface containing droplets evaporates slower, and the virus persists for up to a few days [18]. Virus adhesion to material surfaces involves several factors, including hydrophobicity/hydrophilicity, electrostatic connections that can vary with pH, viral surface charge, and the isoelectric properties of surface proteins [19].

#### 1.1.3. Evaporation

Evaporation mainly depends on the virus droplet volume, ambient temperature, relative humidity, and material contact angle. Sanghamitro et al. [18] investigated the coronavirus droplet evaporation at two different temperatures (25 and 40 °C). At 25 °C, small-sized droplets evaporated in approximately 6 s, while larger droplets took approximately 27 s to evaporate. As the temperature increased, the virus concentration decreased. The contact angle decreased with increasing droplet spread on the surface of the materials; consequently, a smaller contact angle indicated a shorter evaporation time (Figure 2i–iv). The material surface contained droplets with a spherical cap shape throughout evaporation. The volume (*V*) and contact angle (*θ*) of a spherical cap droplet are expressed as follows: V = 1/6πh (3R^2^ + h^2^) and θ = 2 tan^−1^ (h/R), where *h* and *R* are the droplet height and wetness radius, respectively. The contact angle and evaporation rate are different between the materials (porous and non-porous surfaces) and have different timing intervals. Viruses that are persistent on the material surface mainly depend on adsorption and evaporation. In this article, the section of adsorption and evaporation section explained in detail.

#### 1.1.4. Isoelectric Point

The coronavirus is persistent on the surface of the materials for a few hours to days, depending on the probable adsorption of surface properties. Joonaki et al. [20] explained the interactions between the surface protein and material chemistry of the virus. They identified the relationship between SARS-CoV-2 viral proteins and solid surfaces at various pH values below the isoelectric point. The NH_2_, NH_3_^+^, COOH, and COO^−^ groups of amino acids in the SARS-CoV-2 surface-encapsulated S protein (positive charge) mediate solid adsorption through double electrostatic interactions and hydrogen bonding. Between the ionized surface-active species of the virion and oppositely charged surfaces (hydroxyl-containing surfaces, such as wood, cotton, or paper) (Figure 2iv).

The virus adsorbing on material surfaces is influenced by various factors, such as the virus protein surface charge, stability, size, and steric conformation [21]. Viruses are primarily adsorbed onto surface materials via two key systems. One is by van der Waals forces (mainly mineral surfaces), and the second is electrostatic interactions (charged surfaces in the presence of ions and neutral pH) [22]. Hydrogen bonding is a key function in contact with the surface of materials and the potency of viruses. The viral protein on the surface of materials can be deproteinated through strong O-H•••O bond contact with a carboxylic acid on the virus, particularly at a pH close to 4 [23]. Interaction between SARS-CoV-2 surface proteins and substances (solid materials) depends on pH. pH values below the isoelectric point have an overall positive potential for the SARS-CoV-2 surface, under which the viral protein is less likely to be destroyed. The viral surface contains protonated amine and carboxylate groups and hydroxyl-containing materials (paper, wood, and cloth) connected via hydrogen bonding [20].

## 2. Coronavirus in Environment

Virus particles are transmitted to the environment through the air, water, contaminated materials, and climatic conditions; the 150 nm swine influenza virus enters the environment through the air [24]. Human respiratory coronavirus 229E (HCoV-229E) and feline infectious peritonitis virus (FIPV) can survive in water for up to 10 d, and the Zika virus can also survive in sewage for up to 10 d [25]. Coronavirus is persistent in the surrounding environment based on climatic conditions, wind speed, temperature, and humidity [26,27]. In this situation, SARS-CoV is spread via respiratory particles, the particles are >5–10 μm in diameter and can be spread via droplets (viruses contain inside the droplet). The virus can be transmitted to the surroundings via fomites when people come into contact at distances less than one meter [28]. Several factors are involved in the transmission of SARS-CoV-2 from humans to the environment and vice versa, including humidity, temperature, fomites, and ventilation [9]. Figure 3 shows that coronavirus spreading into the environment mostly involved manmade activities [29], as well as spreading through aerosols [30].

### 2.1. Coronavirus-Contaminated Surface Materials

Coronaviruses have spread from infected to uninfected surfaces [31]. The virus spreads through contaminated surfaces of household appliances, such as doors, bed rails, light switches, walls, ceilings, mirrors, windows, and medical equipment such as patient toilet rooms, X-ray machines, knobs or handles, and hemodialysis [32,33]. Kim et al. [34] evaluated the MERS-CoV outbreak in South Korea and discovered that it spread through contaminated surface materials and air. HCoV-229E maintains its infectivity on various non-biocidal surface matter, as well as stainless steel, glass, silicone rubber, PVC, polytetrafluoroethylene (Teflon; PTFE), and ceramic tiles [35]. This study showed viral survival based on the copper alloy concentration; the virus was destroyed more quickly than in other materials, and Cu/Zn brasses were effective at lower Cu concentrations, which damaged the viral spike protein. Numerous porous (paper and wood) and non-porous (plastic and glass) materials spread the virus from one location to another [35].

#### 2.1.1. Porous Materials

##### Paper

Chin et al. [36] reported that SARS-CoV-2 is persistent on paper surfaces for up to 7 d. The 5 µL of Log_10_ 7.8 in TCID_50_/mL of SARS-CoV-2 was added to the paper surface and kept at room temperature with 65% relative humidity, and the virus was persistent on paper for up to 3 h (log_10_ order reduction by 2.18 ± 0.05 in TCID_50_/mL), and after that, the virus was not detected. Liu et al. [7] tested the persistence of SARS-CoV-2 on paper surfaces for up to 7 d. This investigation was conducted at a temperature range of 25–27 °C and 35% relative humidity, resulting in a decrease of the viral titer by log_10_ order 3.4 in TCID_50_/mL over 2 h.

##### Wood

Chin et al. [36] studied the persistence of SARS-CoV-2 on wood surfaces and found it could remain for up to 1 d. They reported a log_10_ order reduction of 2.47 ± 0.23 over 6 h. Another study indicated that the SARS-CoV-2 titer was reduced by a log_10_ order of 1.7 in TCID_50_/mL, with the virus remaining on wood for up to 4 d [7].

##### Mask

The face mask helps to control the spread of the virus from respiratory droplets released through sneezes or coughs. The World Health Organization (WHO) and Center for Disease Control (CDC) recommend wearing a mask to prevent SARS-CoV-2 [28]. Different types of masks are used for virus spread and control based on virus particle penetration rates, such as the surgical mask (penetration rates ~30–70%) and N95 or FFP2 (~5%) [37]. Commercial face mask fabrics can partially block viral particles, and novel material-based masks can inactivate the spread of the virus. Various types of conjugated mask materials are used to control viruses, such as conjugated polymers and oligomers [38], reduced graphene oxide and silver/copper nanoparticles [39], graphene oxide and polydopamine [40], and copper [41]. The outer layer of the mask is a hydrophobic nonwoven layer (e.g., polypropylene, polyester, or polyaramid) that reduces the risk of exhaling and/or inhaling the virus. The surface chemistry of the polymer is used to control the viral attachment to the surfaces of the materials. Cationic polymers are highly attractive for viral binding [42]. Personal productive equipment (PPE) and the surface of polypropylene have a positive charge and hydrophobic nature, and strongly bind to the surface of SARS-CoV-2. The interaction between the virus spike protein and surface materials is mainly based on electrostatic and hydrophobic interactions, which damage the viral protein and consequently inactivate the virus [43].

#### 2.1.2. Non-Porous Materials

##### Plastic

The persistence of plastics in the environment is based on the viral number and types of plastic; these include PVC, polycarbonate (PC), polypropylene (PP), polyethylene terephthalate (PET), and low-density polyethylene (LDPE). Virus persistence on plastic depends on the temperature and relative humidity, and Doremalen et al. [16] reported that SARS-CoV-1 and SARS-CoV-2 can remain on plastic surfaces for up to 4 d, after which they become less persistent. A further study reviewed the HCoV-229E persistence in polytetrafluoroethylene (Teflon) and PVC at a relative humidity between 30 and 40% at room temperature [35]. Interactions between viral particles on plastic surfaces depend on hydrophobic forces and electrostatic interactions [44]. Some studies have shown that virus persistence on PP and PET is influenced by non-ionic forces and the pH of the medium. Under these conditions, the surface charge of the virus particle is assumed to be neutral.

##### Glass

Chin et al. [36] investigated SARS-CoV-2 and found it to persist on glass surfaces at 22 °C and 65% relative humidity for up to 14 d. Over 4 d, a reduction in log_10_ order titer from 7.8 to 3.8 in TCID_50_/mL was observed. Bonny et al. [45] reported that HCoV-229E (VR-740) was persistent on the glass surface for up to 7 d at 24 °C at 50% relative humidity, and on the seventh day reached 2 × 10^4^ plaque forming unit (PFU). Katja Schilling-Loeffler et al. [46] investigated the persistence of HCoV-229E (100 μL of 2.5 × 10^5^ PFU) on a glass slide surface (Menzel GmbH&Co. KG, Braunschweig, Germany) under both light and dark conditions. They found that under dark conditions, the viral titer reduced by 0.5 in 3 d, with a 2.7 log_10_ reduction in PFU after 14 d. The remaining virus was inactivated after 25 d. In light conditions, the virus persisted at a 3.3 log_10_ reduction in PFU for 3 d, and the remaining virus was inactivated after 14 d. Artificially prepared glass also inactivated the virus. Saeed et al. [47] reported that Polyurethane and a Cu_2_O-coated glass slide were used to inactivate the SARS-CoV-2 (5 µL of 6.2 × 10^7^ TCID_50_/mL). This experiment was performed at 22–23 °C, 60–70% relative humidity, and a TCID_50_ assay with Vero E6 cells (African green monkey kidney epithelial cells). The Cu_2_O/PU-coated glass inactivated 99.9% of the viral titer in 1 h (log_10_ order reduction 2 in TCID_50_/mL) and the uncoated glass reduced the log_10_ order titer to 5.3 in TCID_50_/mL after 3 h.

##### Stainless Steel

Riddell et al. [48] reported the persistence of SARS-CoV-2 on the surface of stainless steel. They artificially infected SARS-CoV-2 (10 µL 3.38 ×10^−5^ TCID_50_/mL) on the surface of cleaned stainless steel, and virus-inoculated coupons were placed in a 50% humidified climate chamber (Memmert HPP110) and these samples were maintained at different temperatures (20 °C, 30 °C, and 40 °C) in dark conditions (avoid light involving virus inactivation). In this study, a maximum log_10_ order of >1 in TCID_50_/mL of the virus persisted for up to 28 d at 20 °C and 50% humidity conditions; however, persistence was decreased with increased temperature (30 °C in 7 d and 40 °C in 24 h), and reduced virus log_10_ order titer to 1.5 and >1 in TCID_50_/mL, respectively. Light plays a role in inactivating the coronavirus. Specifically, light generated by a quartz window (UVB: 1.6 to 0.3 W/m^2^) inactivated SARS-CoV-2 that had contaminated stainless steel surfaces [49]. The authors indicated that quartz window-generated sunlight conditions inactivated the virus more quickly than dark conditions (*p* < 0.0001). The SARS-CoV-2 titer reduced the log_10_ order from 3 to 0.40 TCID_50_/mL in 18 min under 1.6 W/m^2^ UVB conditions, and there was no significant viral titer reduction in 60 min under dark conditions.

##### Copper

Copper has self-disinfecting properties and effectively controls viral persistence on household and common touch surfaces. The copper and copper-alloy surfaces effectively controlled the virus. Van Doremalen et al. [16] reported persistent HCoV on a copper surface for 4–8 h in SARS-CoV-2 and SARS-CoV-1. Warnes et al. [35] reported that the persistence of HCoV-229E on copper surfaces is dependent on copper concentrations, noting that the inactivation rate of HCoV-229E is proportional to the percentage of copper. In this study, 20 µL (10^3^ PFU) of HCoV-229E was artificially added to the surface of brass and copper-nickel, and the virus was inactivated within 60 min. Cu inactivated the virus depending on both Cu (II) and Cu (I) ions. The virus was inactivated faster with Cu (II) than with Cu (I). The Cu ion generated hydroxyl radicals, which effectively inactivated the virus morphology, along with clumping and damaging the surface spike protein. Cu interacts with molecular oxygen to generate superoxide, and consequently generates hydrogen peroxide through the Haber–Weiss reaction, as shown below:(1)$$2Cu^{+}+2O_{2}\left(aq\right)\to 2Cu^{2+}+{2O_{2}}^{-}$$
(2)$${2O_{2}}^{-}+2H^{+}+OH^{-}\to H_{2}O_{2}+O_{2}$$
(3)$$Cu^{+}+H_{2}O_{2}\to Cu^{2+}+OH^{-}+OH$$

### 2.2. Environmental Factors Involved in Coronavirus Inactivity

#### 2.2.1. Role of Temperature

SARS-CoV was inactivated under various temperatures (56, 65, and 75 °C) over a range of times (0 to 90 min). A temperature of 65 °C resulted in a log order reduction of 1 within 4 min, while 56 °C achieved the same reduction in 20 min [50]. The survival of viruses in different types of raw materials depends entirely on the temperature and duration (Table 1).

#### 2.2.2. Role of pH

The persistence of coronaviruses observed on surface materials mainly depends on surface chemistry and environmental conditions; the virus attaches to hydroxyl-containing surfaces through hydrogen bonding. The virus link surface depends on pH, the surface material presence of -O–H•••O bonds having a pH above 4, and most of the viral particles negatively charge isoelectric points above pH 7 [58]. The acidic pH of the surroundings is preferred for the blending and dispersion of viruses, and increasing the extracellular pH is the main factor in inactivating SARS-CoV-2. A high-pH level is associated with virus transmission from humans to the environment. High levels of negative air ions (NAIs), including O^−^, O_2_^−^, O_3_^−^, CO_3_^−^, NO_2_^−^, NO_3_^−^, HCO_3_^−^, and OH^−^, can function as Brønsted–Lowry bases by accepting protons. Among these, superoxide ions (O_2_^−^) are most commonly generated in the environment [59].

#### 2.2.3. Role of Humidity

Humidity and temperature collectively inactivate coronaviruses and facilitate the transmission of viral particles, whereas conflicting weather conditions can prolong viral stability on inanimate surfaces. Chan et al. [60] reported that 10 μL (10^−5^ TCID_50_) of serially diluted SARS-CoV (HKU39849) was placed in a non-porous plastic material and dried. Subsequently, the dried plates were incubated at various relative humidity (RH) (>95%, 80–89%), heat (38 °C, 33 °C, 28 °C), and different timing intervals from 3 to 24 h, and residual viral contamination was examined at a monolayer of FRhK-4 cells. The higher humidity (>95%) with low temperatures (28 °C and 33 °C) did not significantly influence the infectivity of the virus and they reduced titer by only log_10_ order 1 at 80–90% humidity and 0.25–2 log_10_ reduction was observed at 38 °C over a 24 h period. Another study also tested SARS-CoV infectivity on stainless steel. They cut the stainless steel into 1 cm^2^ piece and washed it with 0.01% Tween 80, and 70% ethanol, and cleaned it once with sterile distilled water. The cleaned surfaces were put with 10μL of 10^−4^ to 10^−5^ test viruses SARS-CoV, and were kept in different environmental conditions (temperature 4 °C, 20 °C, and 40 °C and RH 20 ± 3%, 50 ± 3%, and 80 ± 3%). The 20% RH reduced the log_10_ order to <0.5 over 28 d, 50% RH was reduced by ~3.5 log_10_ order after 21 d and 80% RH was reduced by log_10_ order 3.2 over 28 d. According to one report, 20% RH attains a low virus reduction compared to 50% and 80% RH [61].

The England-2 variant SARS-CoV-2 was artificially prepared and sprayed using a 3-jet collision nebulizer into a 40 L Goldberg drum controlled by an AeroMP system (BiAera, Hagerstown, MD, USA). The variant SARS-CoV-2 attained a decay rate of 0.91%/min in medium RH (40–60%), and high RH, 68–88%, reached 1.59%/min decay rate, and artificial saliva was more stable at high RH (decay rate of 0.40%/min) than at moderate RH (decay rate of 2.27%/min) [62]. In another study, 5 μL (10^−6.0^) of two types of viruses, namely MERS-CoV (HCoV-EMC/2012) and H1N1 (A/Mexico/4108/2009). were artificially put on a surface of steel or plastic washers (McMaster-Carr, Elmhurst, IL, USA). MERS-CoV that were placed on plastic and steel surfaces were eliminated (virus titer in the log_10_) in 72 h at 40% RH, 20 °C, and MERS was inactivated within 48 h at conditions of 30% humidity at 30 °C. In addition, 80% humidity and 30 °C attained virus inactivation after 24 h. Similarly, H1N1 decreased in viral activity and was not detected after 10 h. These results demonstrate that both increased temperature and RH contribute to reducing viral viability on non-porous surfaces [52]. Other studies have also indicated that the influenza virus spreads most efficiently at an RH of 60–80%, with the lowest efficiency observed at a moderate-range humidity of 40–60% [63].

### 2.3. Virus Inactivation in Chemical and Physical Treatment

Viruses are small in shape, and even fewer viral loads severely contaminate surface materials (porous and non-porous). Coronaviruses are inactivated by chemical methods such as alcohol, chlorine, and peroxide, and physical methods such as temperature, pH, and humidity as environmental parameters, UV irradiation, gamma radiation, X-rays, and non-thermal plasma (NTP)-generated reactive oxygen and nitrogen species (RONS) with good efficacy against the virus. Radiation is used to treat the viral contamination of various types of surface materials with fewer side effects.

#### 2.3.1. Effective Chemicals Control Coronaviruses

Disinfectants are significantly controlled or eliminate pathogenic microorganisms; sanitation strategies include biocidal agents, for example, alcohol, chlorine-based disinfectants, hydrogen peroxide, peroxyacetic acid, formaldehyde, and povidone-iodine [15,64,65]. Chlorine-based sanitizers are mostly dependent on sterilization range, efficiency, low price, and ease of breakdown with slight residue [66]. Several chemicals are used to control the virus on surface-contaminated materials, such as ethanol (70%), bleach solutions (0.1% sodium hypochlorite), and hydrogen peroxide (0.5% H_2_O_2_), along with energetic methods such as heat inactivation, microwave, or UV-based systems [15,67]. Disinfectants (alcohol, hypochlorite, peroxide, etc.) disturb the coronavirus lipid envelope and infectivity of the coronavirus in the environment without causing any side effects. The efficacy of commonly used materials eliminates the infectivity of SARS-CoV-1; these disinfection studies on the initial virus titer of log_10_ order 6.55, using different chemicals with different concentrations of glutaraldehyde (0.5–2.5%), formaldehyde (0.7–1%), 2-propanol (70–100%), a combination of 45% 2-propanol with 30% 1-propanol, and povidone-iodine (0.23–7.5%), chloroxylenol (0.05%), 0.05% chlorhexidine (0.05%) and 0.1% benzalkonium chloride, showed that the log_10_ order reduction ranged from ≥1.68 to ≥5.01 in TCID_50_/mL in the 30 s to 2 min of waiting time at room temperature [68].

##### Alcohols

Ethanol and isopropanol are the main alcohols used to disinfect viruses, bacteria, and fungi, and their biocidal activity is mainly based on their concentration and hydroaffinity. Coronavirus contamination on surfaces was eliminated by different types of chemical disinfectants. Suspension tests for SARS-CoV-1 demonstrated a log_10_ order reduction ranging from >3.9 to ≥5.5 in TCID_50_/mL within 30 s to 10 min when using an ethanol concentration ranging from 70–95% [15]. Alcohol damages the viral membrane and denatures proteins, in addition to damaging RNA. Ethanol can cause skin allergies and eye irritation, whereas expanded usage can cause skin dryness and itching. Ethanol and isopropanol can control surface-containing viruses but are toxic to human health and flammable [69].

##### Surfactants

Surface-active proteins act as surfactants (amphiphilic moieties possessing both lipophilic and hydrophilic segments) and can be divided into anionic, non-ionic, and cationic surfactants. Some anionic surfactants (sodium linear alkyl benzene sulfonate, sodium Laureth Sulfate, and n-lauroylsarcosine) and non-ionic surfactants (Tween-80 (polysorbate-80), Tween-20 (polysorbate-20), and Triton X-100) connect amide, ether, ether–ester, or ester bonds with the virus surface, and denature the protein layer and nucleocapsid protein of the virus [70]. Cationic surfactants, such as quaternary ammonium compounds (QACs), dodecyl dimethyl ammonium chloride, and cetylpyridinium chloride, disturb the lipid and envelope protein layers in SARS-CoV-1, MERS, and SARS-CoV-2.

##### Hypochlorite

Virus inactivation involves chemical oxidation. Chlorine gas reacts with water to produce hypochlorous acid (HOCl), hydrogen ions (H^+^), and chlorine ions (Cl^−^). HOCl further dissociates into hydrogen ions (H^+^), hydroxide ions (OH^−^), and hypochlorite ions (OCl^−^). The chlorine dioxide accepts five electrons, which are thoroughly reduced to Cl^−^ and 2O, as shown in Figure 4. Chlorine dioxide is water-soluble at low temperatures, with a pH range between 5 and 10, and is a powerful oxidative agent that can control the Mg, Fe, odor, and taste of water. Highly reactive species, OCl^−^ and HOCl, create physiological damage to the viral membrane to disrupt protein synthesis [71]. Chlorine mainly damages viral proteins and nucleic acids, while chlorine dioxide (ClO_2_) and monochloramine (NH_2_Cl) mainly break the viral capsid [72].

##### Peroxide

Hydrogen peroxide efficacy of SARS-CoV-2 reached log_10_ order reduction of 1.0–1.8 in a work concentration of 1.5–3% after 30 s, and 1.0–1.3% in 15 s after H_2_O_2_ exposure at room temperature [73]. Moreover, H_2_O_2_ solutions recommended an oral rinse at 1.5%, and 3.0% concentrations show minimal virucidal effect at 30 s [73]. HCoV-229E reduced the log_10_ order to ≥4.00 efficacy after 60 h of exposure to 0.5% H_2_O_2_ [74]. Peroxide-type disinfectants, such as peracetic acid and hydrogen peroxide, oxidize the thiol groups, disulfide bonds of proteins, and even peroxy-produced hydroxyl radicals, which attack different parts of the virus, including proteins, lipid membranes, and nucleic acids [75].

Halogenated compounds povidone-iodine consist of water-soluble povidone-iodine (PVP-I), which effectively inactivates coronavirus. This iodine penetrates the cell membrane and attacks the protein simultaneously as the disulfide bond damages nucleic acid in SARS-CoV-1 at the iodine concentration of 1% or less [76]. The PVP-I can inactivate SARS-CoV-2 on different surfaces [36], and the 0.5% concentration of PVP-I reduces log_10_ order ≥ 4 in SARS-CoV-2 viral titers (inactivation of ≥99.99% of virus) within 15 s [77], and 0.23–7.5% of PVP-I reduced log_10_ order 4.6 in 15–60 s [78]. For formaldehyde and glutaraldehyde, concentrations of 2.5% and 0.5% glutaraldehyde were found to inactivate SARS-CoV-1 viruses within 5 and 2 min, respectively. Additionally, 0.7–1.0% of formaldehyde could reduce the virus titer by >3.0 over 3 min [78].

#### 2.3.2. Physical Methods

##### Ultraviolet (UV)

The UV spectrum is divided into four segments, the first one is vacuum ultraviolet (VUV) radiation (UV radiation 100–200 nm), which is not used for disinfection because of maximum uptake by air, and the other spectrum is UVC (200–280 nm), UVB (280–315 nm), and UVA (315–380 nm), which have strong antiviral and anti-bacterial activity [79]. The UVC-generated light, at 222 nm filtered excimer lamps, inactivates the virus without damaging human cells and tissues [80,81]. Viral inactivation depends on the time and dose. Buonanno Manuela et al. [82] reported that UVC light (207–222 nm) controlled alpha HCoV-229E and beta HCoV-OC43 (Organ Culture 43) at the regulatory exposure limit (~3 mJ/cm^2^/h). The results demonstrated that treatments of 8, 11, 16, and 25 min resulted in viral inactivation efficacy of 90%, 95%, 99%, and 99.9%, respectively.

Darnell Miriam et al. [52] studied the inactivation of coronaviruses using UVA and UVC discharge light; UVA (365 nm) discharged 2133 µW/cm^2^, and UVC (254 nm) discharged 4016 µW/cm^2^ (where µW = 10^−6^ J/s). In this study, African green monkey kidney cells (Vero E6) were infected with SARS-CoV-1 (10^6.33^ TCID_50_/mL), and the virus was inactivated using a UV source at different time intervals for up to 15 min. The UVA did not decrease viral load but UVC reduced virus load 400-fold (log_10_ order reduction ≤ 1.0 TCID_50_/mL). UVC rays are more efficient than UVB and UVA rays in inactivating viral DNA [83]. Matsuura et al. [84] explored the use of a light-emitting diode (LED) light (405 nm) to control a SARS-CoV-2 contaminated titanium dioxide-coated glass fiber sheet (3 cm^2^). In this experiment, they used different photocatalytic reaction timing intervals (0–120 min), and 120 min exhibited more viral inactivation and less virion RNA compared with the control (10^2.63^ TCID_50_/mL). UVC lamps pose a higher risk to the nose, throat, and lungs, mainly for those with respiratory sensitivities such as asthma or allergies [85].

##### Gamma Radiation

Gamma radiation is the ionizing energy or irradiation that interrupts processes, which leads to inactivating the microorganism dosage of 0.2 to 25 Gray (Gy). A dose greater than 10 Gy of gamma radiation inactivates the viruses, and a dose of 25 to 40 Gy [1 Gy is equivalent to the absorption of 1 J/kg] sterilizes the medical device. Gamma irradiation, directly and indirectly, inactivates viruses. Direct gamma irradiation damages fewer viral envelope proteins and is caused by the radiolytic cleavage or rearrangement of the genetic material [86]. This irradiation penetrates and directly damages the nucleic acid without disturbing the structural proteins [87]. The rotating type of the JL Shepherd Model 484R generated gamma irradiation with a cobalt-60 source and was used to control the SARS-CoV-1 (1 × 10^6^ in TCID_50_/mL) and other viruses within 0.5 megarad (Mrads; 1 rad = 0.01 Gy) [88]. Zaire ebolavirus rVSV-EBOVgp-GFP [recombinant vesicular stomatitis virus (rVSV), which encodes the *Zaire ebolavirus* (EBOV) glycoprotein (GP) in place of the VSV GP and contains an additional transcription unit encoding a green fluorescent protein (rVSV-EBOVgp-GFP)] is completely inactivated by gamma (cobalt-60 source) irradiation. The 3 Mrad dosage of gamma radiation inactivates the viruses by 90% (log_10_ order reduction 1 in TCID_50_/mL), and reduced infectivity would be achieved as a sterility assurance level (SAL) of 8.51 × 10^−6^ in rVSV-EBOVgp-GFP, 3.20 × 10^−6^ for La Crosse virus (LACV), and 1.39 × 10^−6^ in rMVKSEGFP(3). Recombinant Measles virus, the Khartoum–Sudan strain expressing an enhanced green fluorescent protein (EGFP), is based on a wild-type genotype B3 virus [89]. Different types of virus inactivation are mainly based on the genome size with-irradiated Mrads; the coronavirus protein ~29 kb was inactivated in 2 Mrads, filoviruses (~19 kb) in 4 Mrads, arenaviruses, orthomyxoviruses (~13 kb) in 8 Mrads, and flaviviruses (~9 kb) in 4 Mrads [88]. The viral inactivation is mainly based on Mrads doses (less than 0.5 Mrads doses did not effectively control the virus), irradiation temperature, sample composition, and distance between the sample and irradiation source [89].

##### X-rays

X-rays generate highly ionizing radiated photons that effectively eliminate different types of viral families, including Phenuiviridae, Nairoviridae, Togaviridae, and Flaviviridae. Lower-energy X-rays control biological contamination more efficiently than high-energy X-rays; however, soft X-rays (1–10 keV) inactivate the virus in both enveloped and nonenveloped proteins [90]. X-ray irradiation is an alternative to chemical and thermal approaches for viral control. The possibility of a photon interacting with a given atom generally decreases with increasing energy; however, the photon penetration decreases with decreasing energy.

##### Ozone

Ozone is a dominant oxidizing mediator that generates singlet oxygen and hydroxyl radicals to inactivate viruses [91]. Some studies used O_3_ to inactivate viruses in personal productive equipment (PPE); Clavo et al. [92] worked to inactivate SARS-CoV-2 contaminated hospital protective and face masks under different conditions, such as time (30 s–10 min) and relative humidity (53–65%) with various mixed concentrations of ozone O_3_/O_2_ combinations of 500 to 40,000 ppm (1–80 g/m^3^). In this study, 4000–10,000 ppm of O_3_ concentration (8–20 g/m^3^) enhanced virucidal activity within 1/2 to a few minutes, whereas 4–6 ppm of O_3_ concentrations (0.008–0.013 g/m^3^) eliminated the viruses in 30 min. Yano et al. [93] studied the inactivate SARS-CoV-2 (50 µL, 8.5 × 10^5^ PFU) on the surface of contaminated stainless steel (3 cm^2^ area) using two different ozone (TM-04OZ; Tamura TECO Ltd., Higashiosaka, Osaka, Japan) conditions, in which the SARS-CoV-2 inactivation attained log_10_ order reduction 3.3 in 1 ppm for 60 min (concentration-time value 60) and 6 ppm for 55 min (CT value 330). This experiment was performed at 25 °C and an RH of 60–80%. DBD plasma (ozone concentration 120 ppm with power 13 W) generated ozone gases, which could eliminate the surface contaminated HCoV-229E on Korean face mask-94 (KF94) under different operating times of 10–300 s. They could reduce the viral RNA load to 4 log orders when the ozone concentration of 120 ppm has been exposed by 10 s [94]. Low concentrations of ozone are non-toxic, but high concentrations induce toxicity in humans [95].

##### Non-Thermal Plasma (NTP)

Recently, we have observed increasing interest in the prevention and control of the SARS-CoV-2 using non-thermal plasma (NTP). Plasma is a moderately or completely ionized gas with several reactive species (free radicals, ions, electrons, and neutral molecules). Ionized gas contains a complex composition of RONS, and this approach effectively controls viral contamination. Plasmas can be created using various electrode geometries, feed gases, gas temperatures, excitation voltages, plasma powers, dominant reactive species, and gas-residence times. All of these contain plasma factors that affect the virus. The virus is significantly inactivated by broad substrates, such as solutions, surfaces, and tissues in two types of plasma devices: dielectric barrier discharge (DBD) plasma and jet plasma [96,97]. The surface-type plasma device consists of two parallel silver electrodes printed on the same insulating substrate. The two electrodes were separated by a 100 to 200 µm and 3–5 µm thickness and these are sealed with an insulating SiO_2_ paste, as shown in Figure 5A. The jet plasma consists of a power-needle electrode, a grounded electrode, a quartz tube, and a high-voltage power supply. The conical outer electrode is fabricated with stainless steel, and the center of the electrode has a 1 mm hole, through which ambient air is ejected into the surroundings. The distance between the needle tip and the grounded electrode is 2 mm, and the plasma jet plume is produced approximately 5 mm from the nozzle, as shown in Figure 5B [94].

The plasma reaction contains over 80 species in humid air, which can generate highly reactive RONS, which inactivate pathogens [98,99]. RONS, including O, ^1^O_2_, O_3_, OH, HO_2_/O_2_^−^, H_2_O_2_, HNO_2_, NO, NO_2_^−^, ONOOH/ONOO^−^, and N_2_O_5_, inactivate pathogens [100,101,102]. Neha et al. [97] reported that photodynamic therapy, X-rays, gamma radiation, and UV radiation-generated ROS control virulence using a cold atmospheric plasma method. Plasma considerably reduces human adenovirus infectivity and replication ability [103], degrades purified SARS-CoV-2 RNA, and prevents the spike protein from attaching to the host cell [104,105]. Few studies indicated that nonthermal plasma controls water- and material-surface-contaminated viruses.

Zhhitong et al. [106] reported an argon and helium (6.4 L/min and 16.5 L/min, respectively) atmospheric pressure plasma jet (APPJ), by which surfaces contaminated with SARS-CoV-2 could be inactivated by RONS on plastic, metal, cardboard, football, and baseball. In this study, material surfaces were infected with SARS-CoV-2 (25 μL of 2 × 10^5^ PFU/mL) and then treated with plasma for durations of 0, 30, 60, and 180 s. SARS-CoV-2 was found to be inactivated on metal surfaces within 30 s. Viruses contaminated on plastic, leather, and football surfaces, as well as cardboard, can be inactivated within a timeframe of 30 to 60 s. Cardboard and basketball surfaces exhibit effective virus inactivation for a 60 s treatment, while few data points exhibited these effects after 30 s of treatments, as shown in Figure 6A. These results suggest that virus persistence is dependent on three important aspects of surface inactivation of SARS-CoV-2: cold atmospheric pressure plasma (CAP) material composition, roughness, and absorptivity. The higher discharge intensity generated by RONS highly inactivated the virus in metals compared with other materials.

Lee et al. [107] artificially put 3.9 TCID_50_/mL of HCoV-229E over a surface of a Korean filter 94 (KF94) mask (Figure 6B). In this study, ozone was produced from the DBD plasma with a 1 mm-thick alumina dielectric plate between the two electrodes to inactivate the mask surface containing the virus. The contaminated mask was treated with DBD plasma-generated ozone for different time intervals (10–300 s). A concentration of 120 ppm ozone inactivated the virus within 60 s, after which it was not detectable. Even when the treatment time was increased for masks, the RT-PCR cycle threshold (Ct) values increased from 24.5 ± 0.2 at 0 s to 28.3 ± 0.2 at a treatment time of 300 s. As the ozone treatment time increased, the Ct values of HCoV-229E recovered from the masks also increased, indicating a decrease in the amount of intact viral genome.

Pradeep et al. [108] investigated a 2 L/min air-fed DBD (frequency of 30 kHz and voltage of 12 kV) plasma-generated, ozone-inactivating HCoV-229E. In this study, a benchtop chamber (500 mm × 600 mm × 600 mm) was used to manage surface virus contamination. They artificially put 100 µL of the virus on a covered glass surface and inactivated it at different ozone concentrations (0–80 ppm) with a treatment timing of 2–4 h. In this study, an increase in ozone concentration reduced the cytopathic effect (Figure 7d–h), with maximum virus inactivation achieved at an ozone concentration of 1 ppm over 4 h (Figure 7a–c). The cytopathic effect (CPE) was directly related to the virus titer, and accordingly reflected the immunofluorescence and HCoV-229E N gene expression, as shown in Figure 7i. Plasma-generated ozone (PGO) reduced the virus log_10_ order 2 and 3.4 in TCID_50_/mL under ozone concentrations of 10 ppm and 80 ppm, respectively. Plasma showed 99–99.9% inhibition of HCoV-229E compared with the control groups, and the selective index was defined as the ratio of the 50% cytotoxic concentration (CC_50_) to the 50% antiviral concentration (IC_50_). In the present study, the SI was 6.51.

Yamashiro et al. [109] reported Feline calicivirus (FCV F9: 20 µL of 0.1 MOI (Multiplicity of infection)) contaminated the cover glass surface. The glass surface containing the virus was treated with DBD plasma (10 kV peak-to-peak, 10 kHz, airflow 3.5 L/min). The plasma reduces the virus titer from 3.81 × 10^4^ ± 1.58 × 10^3^ to 2.11 × 10^2^ ± 6.25 × 10^1^ TCID_50_/mL after 1 min and did not detect the virus in 2 min. It was found that 4mM of plasma-generated ONOO^−^ reduces the virus titer to 4.87 × 10^0^ ± 1.06 × 10^0^ in TCID_50_/mL, and 3% of H_2_O_2_ attains 1.88 × 10^4^ ± 6.57 × 10^3^ in TCID_50_/mL. However, the number of radical scavengers increased RNA damage (Figure 8A). DBD plasma was utilized to inactivate aerosols containing *Staphylococcus epidermidis* and SARS-CoV-2 RNA. It resulted in a reduction of log_10_ order by 3.76 CFU/mL in *S. epidermidis* and a viral RNA-dependent RNA polymerase (RdRP) threshold (Ct) value of 19.33 in the positive control. SARS-CoV-2 RNA was not detectable after 150 s of plasma treatment (Figure 8B) [104]. Nonthermal plasma not only inactivates surface disinfection, but also affects the efficacy of liquid-contaminated viruses. A plasma-activated solution (PAW) inactivates the virus. Guo et al. [102] reported that an air-fed surface discharged plasma device [discharge power density 0.25 W/cm^2^ (23 kHz), voltage 8.36 kV (572.4 mA)] simulated PAW removes the SARS-CoV-2 S protein receptor binding domain (RBD) variant. In contrast, 10 min of DBD-treated PAW generated long-lived H_2_O_2_, NO_2_^–^, NO_3_^–^, short-lived species ONOO^–^, and O_2_^•^ were involved in aggregation (45–65 kDa) and fragmentation (~28 kDa) in the RBD (Figure 8B).

Qin et al. [110] used pulse power-driven, CAP-generated, short-lived reactive species of ONOO^−^, O_2_^−^, ^1^O_2,_ and OH^−^ to inactivate SARS-CoV-2 RBD in the spike protein. Pseudotyped SARS-CoV-2 S protein variants were prepared and used for the virus inactivation study. The protocol is shown in Figure 9A. An argon-fed plasma jet was used to inactivate the virus; a flowchart is shown in Figure 9B. Vero E6 cells were grown in 96 well plates in DMEM containing 10% FBS medium and the cells were infected with SARS-CoV-2-related pangolin coronavirus GX_P2V (2 × 10^5^ PFU/mL), and then treated with Ar-fed CAP at different time intervals (10–300 s). The result indicated that a 10 s CAP exposure time initiated virus inactivation (28.66%) and 300 s (99.94%) almost completely inhibited the virus. The CAP-generated, short lifespan RONS species were proven to be four major species of ONOO^−^, O_2_^−^, ^1^O_2,_ and ^⋅^OH for inactivation of the virus (Figure 9C–F). Furthermore, the 180 s CAP treatment destroyed the structure of spike (S) protein particles, revealing shrunken and rough spherical structures. Major RONS, like ONOO^−^ and O_2_^−^, are known to oxidize the tryptophan, tyrosine, and histidine situated at RBD and NTD (nucleotide triphosphate), which may damage the connection capability of RBD to cell receptor ACE2 and the function of NTD (Figure 9).

Nonthermal plasma not only inactivates surface-contaminated coronaviruses, but also treats other coronaviruses in water. Guo et al. [111] inactivated bacteriophage suspensions using cold atmospheric surface discharge plasma-generated RONS. Water-contaminated bacteriophages were treated in the liquid using surface-type plasma; 4 lpm of a mixed gas of argon (99%) and air (1%) was used (Figure 10A). Plasma-induced aqueous RONS levels were measured at 1 and 2 min of treatment, where surface discharge plasma generated long-lived H_2_O_2_, NO_2_^−^, and NO_3_^−^_,_ whose concentrations were measured to be 221, 8, and 216 µM, respectively, over a 2 min treatment time (Figure 10B). The short-lived RONS species were recognized and measured by electron spin resonance (ESR) spectroscopy using four spin traps: 5,5-dimethyl-1-pyrroline N-oxide (DMPO) for OH trapping, 2,2,6,6-tetramethylpiperidine 1-oxyl (TEMP) for ^1^O_2_ trapping, *N*-(dithiocarbamoyl)-*N*-methyl-d-glucamine (MGD) for NO trapping, and 1-hydroxy-2,2,6,6-tetramethyl-4-oxo-piperidine (TEMPONE-H) trapping for O_2_˙^−^NO_2_˙, and ONOO^−^. After 2 min of plasma treatment, specific short-lived RONS concentrations were observed in the spin adducts. The spin adduct concentrations of DMPO-OH, 2,2,6,6-tetramethylpiperidine 1-oxyl (TEMPO), nitrosyl-Fe, and 4-oxo-2,2,6,6-tetramethylpiperidine-1-oxyl (TEMPONE) were 0.5, 145, 30, and 188 μM, respectively, after 2 min of plasma treatment (Figure 10B).

Plasma-activated water was used to control bacteriophage contamination in the aqueous solutions. The four different conditions used to inactivate the bacteriophage are as follows: (1) without treatment (control); (2) T4 bacteriophage plasma treated at 60 s, Φ174 and MS2 bacteriophage plasma treated at 40 s; (3) long-live species mixture in 500 µM of H_2_O_2_, 75 µM NO_2_^−^, 500 µM NO_3_^−^; and (4) 1% formaldehyde. After plasma treatment, the bacteriophages were incubated at 22 °C for 8 h. The T4 bacteriophage reduced 8.8 PFU/mL in 8 h (Figure 10C), Φ174 and MS2 reduced 124 and 21 PFU/mL in 6 h and 4 h, respectively (Figure 10D,E), and also 1% formaldehyde reduced less than 2 PFU/mL in 8 h. The 100 s plasma-activated water was stored at 22 °C for 10 d. Every 3 d, the plasma-activated water was removed and mixed with T4 bacteriophage suspensions at a 1:1 ratio. The stored plasma-activated water exhibited weak antiviral activity (<4.5 PFU/mL) because of the decay or reduction of reactive species during long-term storage (Figure 10F).

### 2.4. Virus Persistence and Inactivation Mechanisms

Three factors are involved in viral persistence on the surface of materials: (1) porosity; (2) environmental factors (temperature, relative humidity, and material surface); and (3) absorption [20]. The viruses can persist for 3–4 d on porous materials (paper, wood, and masks) and for longer than 7–19 d on non-porous materials such as plastic, glass, and stainless steel. Viruses persist on materials depending on environmental factors and material characteristics. The contact angle was one of the factors affecting the persistence of the virus on the surface of the material; porous materials evaporate water droplets more quickly than non-porous materials. The paper contact angle was 0°, and the stainless steel, plastic, and glass contact angles were >70° after 45 min (unpublished data). Viruses persisted for longer periods in non-porous materials because these non-porous conditions do not allow full dryness of the surface of a material. Microdroplet evaporation conditions and virus persistence in the materials were correlated (unpublished data). The persistence of viruses in different types of materials was entirely dependent on temperature and duration (Table 1). Other parameters, such as RH and low humidity (20%), attain low virus reduction compared to 50% and 80% RH [61].

Hydrophilic (water-attractive or water-loving) and hydrophobic (water-hating or repelling) natures are the main parameters for viral persistence/inactivation. Hydrophilic materials (paper, mask, wood, stainless steel, and Cu) are polar and easily form hydrogen bonds with material surfaces containing viruses; the contact angle (θ) is less than 90°. Hydrophobic materials (cover glass, and plastic; contact angle (θ) is greater than >90°) are nonpolar materials with a low affinity for water, which makes them repel water. The inactivation of viruses is crucial, and the antiviral properties of surface materials may be directly related to their ability to absorb viruses. In the ambient environment, virus inactivation is determined by electrostatic interactions, such as pH, isoelectric point (pI), ionic strength, hydrophobic effects, and small-scale non-covalent bonds (e.g., van der Waals forces), all of which control the binding of the SARS-CoV-2 spike protein to solid surfaces [112]. All these parameters are essential for hydrophilic/hydrophobic characteristics. Water molecules have strong polarity, and the surface of the viral protein (carboxylate and amine groups) is connected to the surrounding water through hydrogen bonding (Figure 2iv). This binding promotes viral protein hydration and strong viral protein binding to hydrophilic materials (i.e., paper, wood, mask, and stainless steel), and the water molecules can fill the space between the virus particles. In hydrophobic surface materials (glass and plastic), the virus interaction is weak, and the round expansion of the water drop is excessively small to yield an even inter-gap between virus particles and materials. Thus, hydrophilic materials have greater viral inactivation than hydrophobic materials.

Surface-contaminated viruses are inactivated by numerous chemical reagents, and the chemical compounds used as decontaminators are not only harmful to human health but also affect animals and aquatic ecosystems. The nonthermal plasma-generated RONS effectively controls the virus without damaging the materials and is safe for the environment. Nonthermal atmospheric pressure biocompatible plasma (NBP), particularly some types of plasma devices such as facing-type discharged plasma and dielectric barrier jet plasma, are used in bioscience and medicine. This plasma-generated electron temperature ranges from 0.8–3 eV and plasma densities are 3 to 5 × 10^12−16^ cm^−3^ [113]. This plasma starts UV photolysis in the liquid to generate RONS and inactivates pathogenic microorganisms. This review article also reports different types of plasma sources, such as micro-DBD, jet plasma, and nanosecond discharged plasma, used to inactivate pathogenic microorganisms, particularly coronaviruses.

Several plasma studies have focused on inactivating the virus; surface-type DBD controlled the virus more effectively than other plasma devices. DBD plasma-generated reactive nitrogen species (RNS) effectively disinfected different types of porous and nonporous materials. In this experiment, surface-type DBD plasma inactivated HCoV-229E in 3.5 LPM air feed and less than 0.1 ppm concentration of ozone. Although less than 0.1 ppm concentrations of O_3_ required long time intervals for effective HCoV-229E inactivation, porous materials containing viruses are inactivated more effectively than nonporous materials. The virus was inactivated with or without plasma treatment for up to 5 h. A viable virus was detected up to 5 h, the virus titer was from 4.50 to 1.17 TCID_50_/mL without plasma treatment and 4.50 to 0.00 (nondetectable) TCID_50_/mL in plasma (P) treatment. Nonporous materials reduced the virus from 4.77 to 1.17 TCID_50_/mL in plastic and stainless steel except Cu. Compared to all other materials, copper has greater viral inactivation because it reacts with water droplets to form CuO and more RONS. Consequently, the droplets evaporate, and copper produces a high amount of RONS, leading to 99.9% virus inaction within a short period. Plasma treatment effectively controlled the virus compared to that without plasma treatment (unpublished data).

The antiviral mechanism of materials is mostly dependent on the contact between viruses and materials and the formation of RONS by non-thermal plasma (Figure 11). RONS (H_2_O_2_, OH, NOx, and O_2−_) are generated by nonthermal plasma, which damages viral proteins and RNA (Figure 11A–C). Guo et al. [102] reported that 5 and 10 min of plasma-activated water inactivated the SARS-CoV-2 spike protein of the pseudovirus, and the RBD protein was more damaged at 10 min. Tu et al. [114] reported the use of porous nickel (Ni) to inactivate SARS-CoV-2 via electrochemical oxidation using NiOOH (cathode) and Na_2_CO_3_ (anode) in an aqueous solution. They reported a higher viral inactivation rate of 99% in 2 min and 95% in 30 s, and the electrochemical interactions also damaged the viral RBD.

NO is a potent molecule that binds to intracellular molecules and is crucial in various biological functions. The oxidation of NO to l-arginine and l-citrulline is catalyzed by three enzymes. Enzymes include neuronal nitric oxide synthase (nNOS), inducible NOS (iNOS), and endothelial NOS (eNOS) [115]. iNOS/NOS2 is an enzyme initially expressed by activated macrophages and generates NO from the amino acid l-arginine, thereby helping to control the replication or killing of intracellular pathogens. NO involves a broad range of cellular signaling molecules and has an inhibitory effect on viral infections. Akerström et al. [116] reported that the NO donor S-nitroso-N-acetylpenicillamine (SNAP) significantly inhibited SARS-CoV replication compared to that without the NO donor N-acetyl penicillamine (NAP). They showed that iNOS and virus addition reduced the SARS-CoV titer from 2.1 × 10^7^ to 3.9 × 10^6^ TCID_50_ in Vero E6 cells and also indicated the inhibition of SARS-CoV RNA replication.

## 3. Conclusions

SARS-CoV-1 and SARS-CoV-2 demonstrated high infection rates (these viruses encoded more accessory proteins and created more sgRNAs (single guide RNA)) compared with older strains of coronaviruses, such as HCoV-OC43 and HCoV-NL63 (Netherlands 63). Viruses persist on surface materials through two mechanisms: (i) van der Waals interactions (mostly mineral surfaces/solid materials) and electrostatic interactions (charged surfaces in the presence of ions), and (ii) temperature and humidity [21]. To prevent viral infection in humans, frequent cleaning and disinfection of environmental surfaces is essential. While chemical disinfectants help control the virus in the environment, they also contribute to environmental pollution. Moving forward, we must develop a method to remove or inactivate contaminating viruses from surfaces using environmentally friendly materials. Non-thermal plasma treatment is an environmentally friendly approach that can effectively control surface-contaminated viruses by producing RONS. Virus inactivation is mainly based on the timing of the pulsed high voltages and electrical discharges used to treat the virus. Currently, we must develop specific remedies against the coronaviruses’ surface materials contaminated with coronaviruses.

The surface contains virus removal (inactivation) based on NTP-generated ROS, RNS, and UV; together, they synergistically damage the viral protein and genomic materials. Controlling the virus depends on plasma-generated reactive species, particularly long- and short-lived reactive species. The main ROS are superoxide radicals (O_2_^−^), singlet oxygen (^1^O_2_), and ozone (O_3_). The NTP reacts with water to form hydrogen peroxide (H_2_O_2_) and hydroxide radical (^⋅^OH). Plasma-generated RNS, including NO, NO_2_^−^, N_2_O, N_2_O_5_, NO_3_^−^, HNO_2_, HNO_3_, NO^⋅^, NO_2_^⋅^, and ONOO, inactivate the virus.

Characterization of materials is crucial for controlling coronavirus, particularly energy-dispersive X-ray spectroscopy, X-ray photoelectron spectroscopy, surface morphology, shape and size of particles, and porous nature. Plasma-generated RONS significantly reduced viral and upcoming outbreaks. Therefore, we must improve plasma combination or synergistic treatment methods. Currently, inadequate information is integrated between the plasma and viruses, and the main problem is creating effective NTP devices to control viruses. However, the potential viral inactivation efficacy is the most important factor. Accurately performed plasma setup systems remain unavailable, and upcoming scientific research is required. NTP-generated RONS inactivate the surface-contaminated viruses of the material. Small- and large-scale NTP equipment should be built without changing RONS production. By overcoming this problem, large amounts of viral contaminants can be inactivated, and the desired plasma strength can be stabilized. The inactivation of the virus using NTP generates numerous species that are tremendously complex. The lethal nature of reactive species makes it difficult to identify their characteristics without interference from others; therefore, investigating the mechanism by which virus inactivation occurs is challenging. This novel catalytic plasma coordination will lead to a new era of virus inactivation in the future.

## Figures and Tables

**Figure 1 ijms-24-14106-f001:**
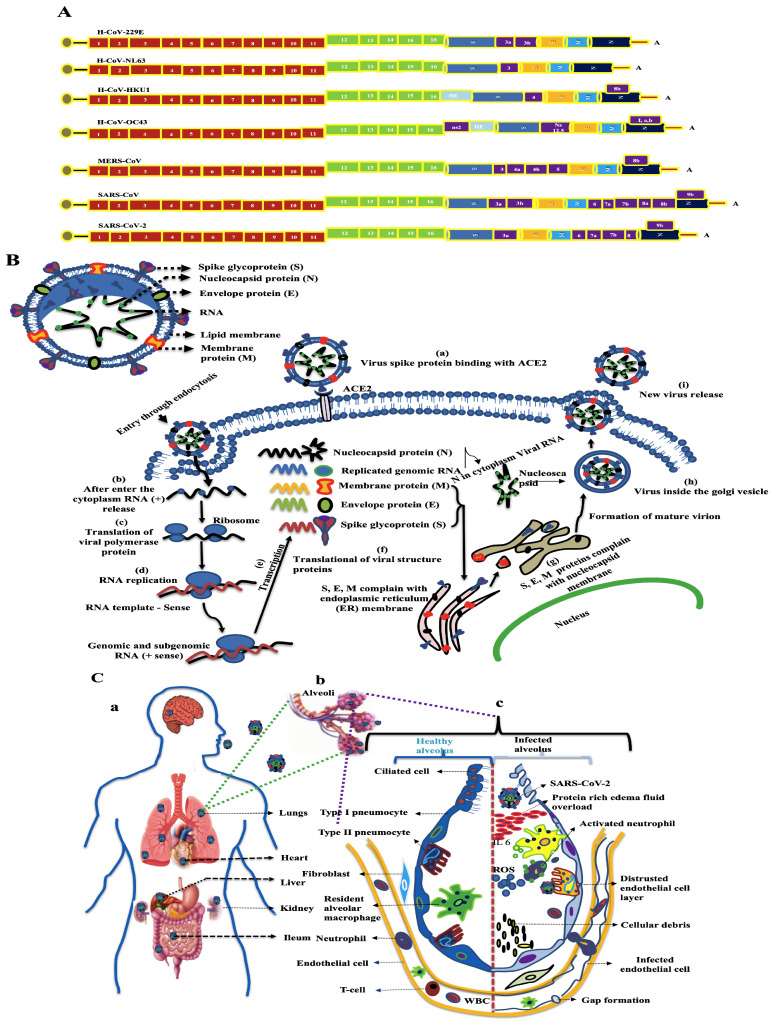
(**A**) Genomic structure of HCoV: Viral genome encoded positive-sense single-stranded RNA, the cap region at the 5′-end (grey circle) and a poly-A as a tail (A30–60) region at 3′ ends. All HCoV contain 1 to 16 nonstructural proteins; out of that 1 to 11 are ORF1a and 12 to 16 are ORF1b, from the left side 4–5 structural proteins are present (S-Spike, E-Envelope, M-Membrane, N-Nucleocapsid, and HE-Hemagglutinin Esterase) along with other nonstructural proteins. (**B**) The life cycle of SARS-CoV-2 in host cell: (a) Virus binding to spike protein ACE2 (Angiotensin-converting enzyme 2), (b) RNA release, (c) part of viral polymerase protein translated, (d) translation of viral polymerase protein, (e) transcription, (f) subgenomic RNA translational S, E, M complain of endoplasmic reticulum (ER) membrane, (g) S, E, M combine with nucleocapsid protein, (h) mature virus inside the Golgi vesicle, and (i) new virus released. (**C**) SARS-CoV-2 infected human organs (a), infected alveoli (b), and healthy and infected alveoli (c).

**Figure 2 ijms-24-14106-f002:**
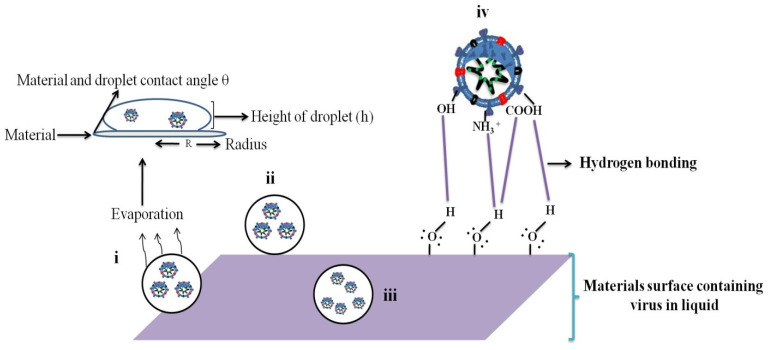
Virus persistence on surface materials: (**i**) Fomite evaporation, (**ii**) High contact angle (hydrophobic materials: plastic and cover glass), (**iii**) fomite fully adsorbed porous materials (hydrophilic materials: paper), (**iv**) virus spike protein contact with materials through hydrogen bonding (viruses are embedded in water and materials).

**Figure 3 ijms-24-14106-f003:**
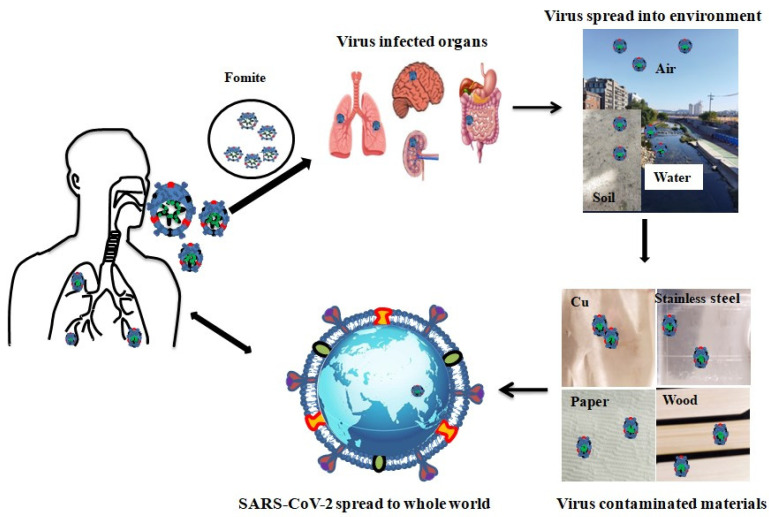
SARS-CoV-2 in infected organs and the Environments.

**Figure 4 ijms-24-14106-f004:**
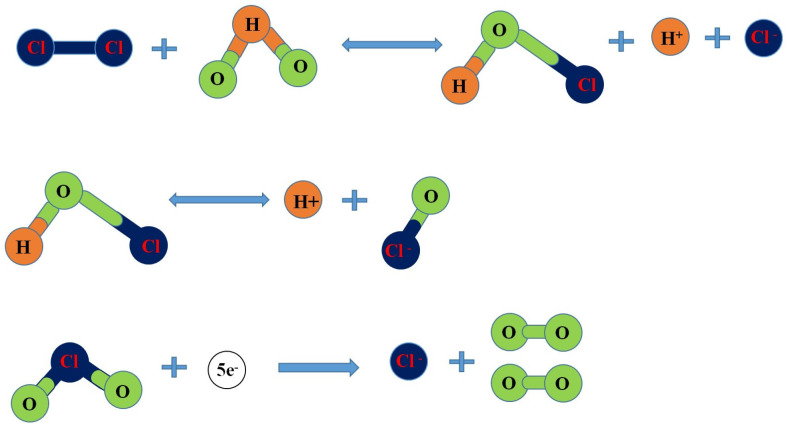
Chlorine reaction in water.

**Figure 5 ijms-24-14106-f005:**
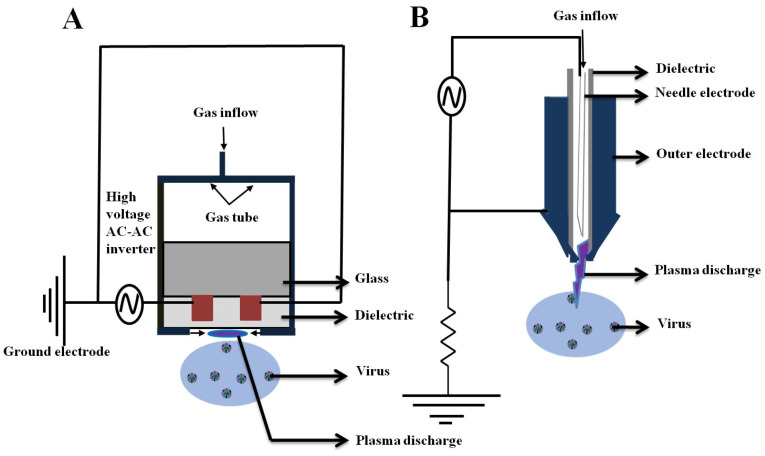
Non-thermal plasma devices: (**A**) μ-DBD surface discharge plasma, and (**B**) Soft plasma Jet.

**Figure 6 ijms-24-14106-f006:**
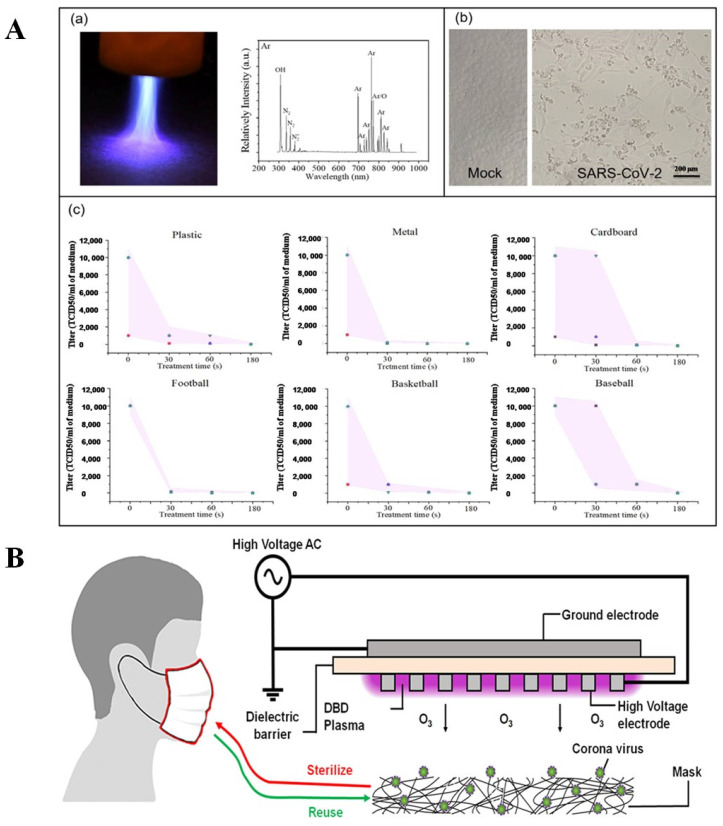
(**A**) Argon-fed, APPJ-generated RONS inactivate the SARS-CoV-2: (a) Plasma treatment in materials and the optical emission spectrum (exposure: 250 ms), (b) Vero-E6 cells indicated SARS-CoV-2 non-infected (Mock) and infected (cytopathic effect) cells. (c) SARS-CoV-2 infected materials CAP treatment at different time intervals (0 s, 30 s, 60 s, and 180 s) on surfaces of plastic, metal, cardboard, football, basketball, and baseball. Reprinted with permission from [106] Copyright 2023AIP publishing Ltd. (**B**) DBD plasma-generated ozone inactivates coronavirus-contaminated masks. Reprinted with permission from [107] Copyright 2023 ACS COVID-19 subset.

**Figure 7 ijms-24-14106-f007:**
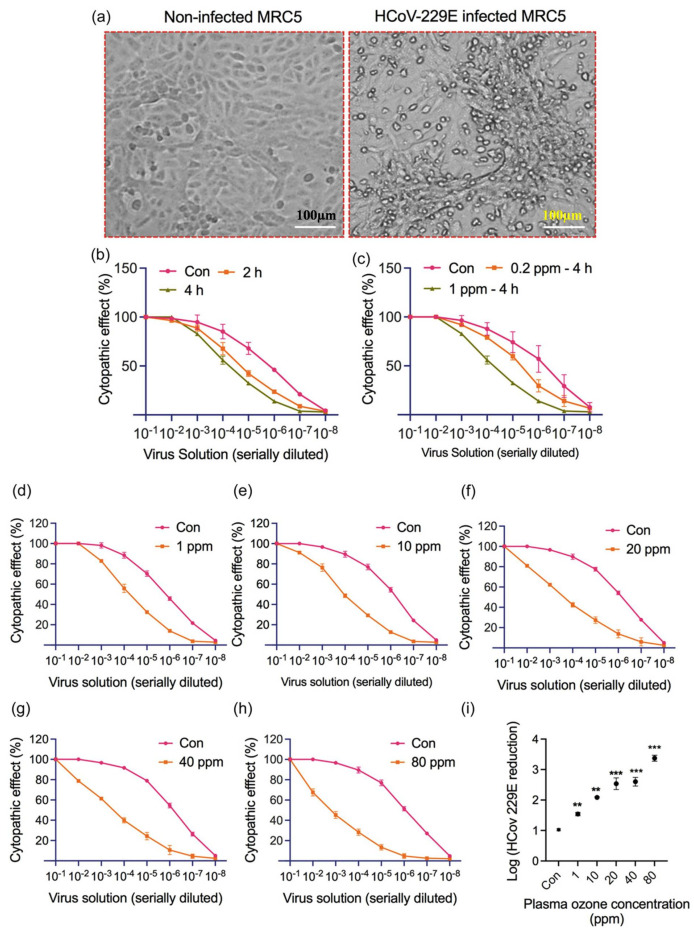
(**a**) HCoV-229E infected and uninfected MRC-5 cell, (**b**,**c**) CPE represents the treatment time and ozone concentration, (**d**–**h**) a percentage of CPE represents the ozone concentrations 1–80 ppm, and (**i**) virus titer reduction based on ozone concentrations. Reprinted with permission from Pradeep et al. [108]. Copyright 2023 John Wiley and Sons Ltd. Statistical significance (n = 2) was calculated and denoted as ** *p* < 0.01, and *** *p* < 0.001.

**Figure 8 ijms-24-14106-f008:**
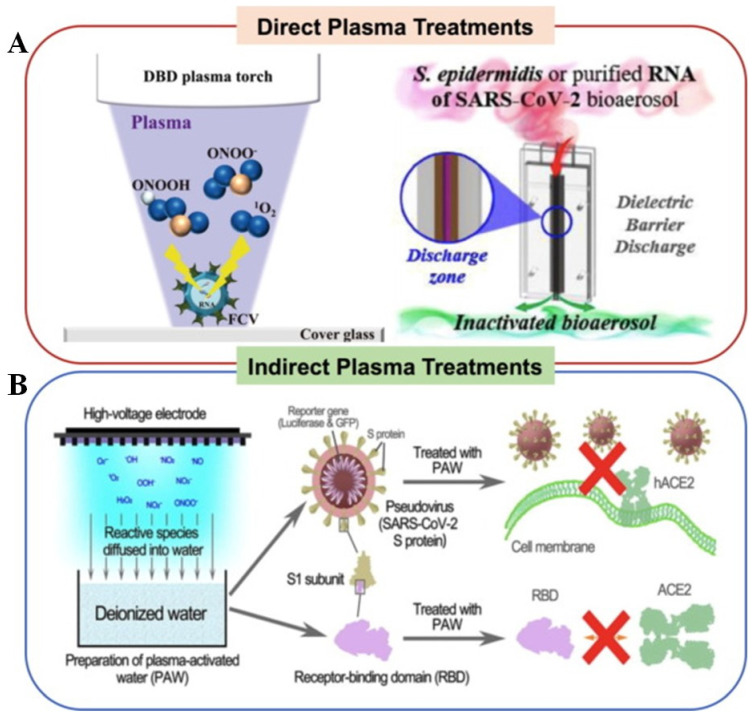
(**A**) DBD plasma torch inactivates FCV (Yamashiro et al., 2018) [109], CAP inactivates aerosol containing *S. epidermidis* and SARS-CoV-2 (Bisag et al., 2020) [104], (**B**) plasma-activated water inactivates the SARS-CoV spike protein (Guo et al., 2021) [102]. Reprinted with permission from [97].Copyright 2023 Elsevier Ltd.

**Figure 9 ijms-24-14106-f009:**
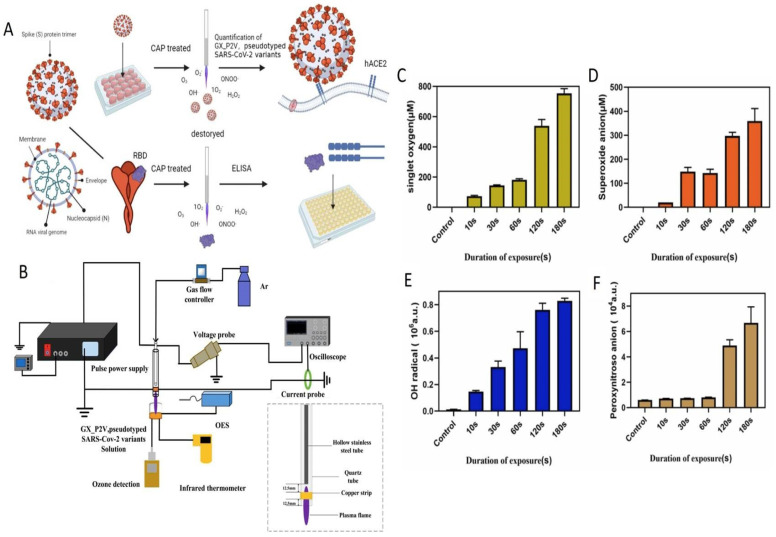
Virus inactivation flow chart (**A**), Plasma jet device (**B**), the plasma produced reactive species: singlet oxygen (^1^O_2_) (**C**), superoxide anion (O_2_^−^) (**D**), hydroxyl radical (⋅OH) (**E**), peroxynitrite anion (ONOO) (**F**), Reprinted with permission from (Qin et al. [110]). Copyright 2023 Elsevier Ltd.

**Figure 10 ijms-24-14106-f010:**
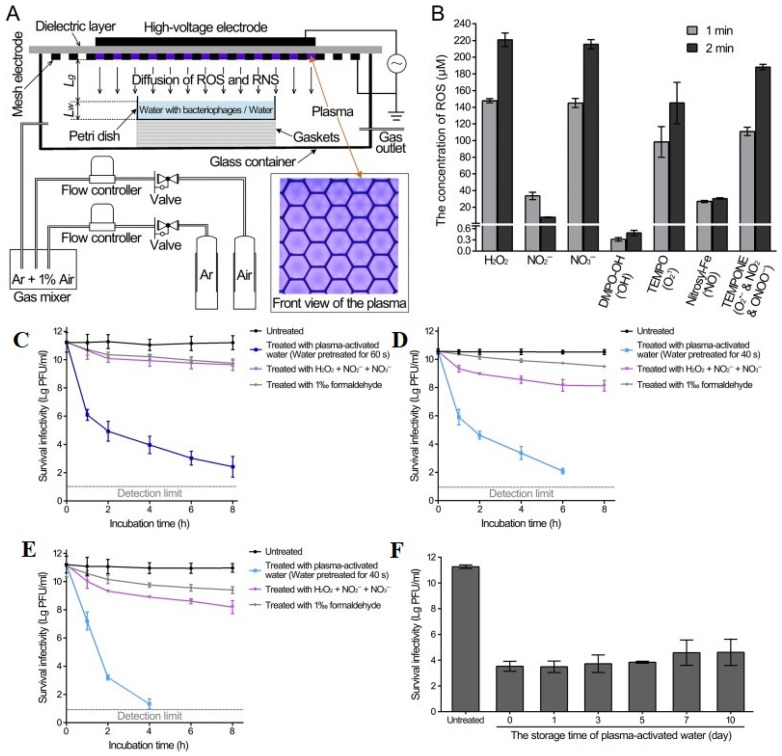
(**A**) Atmospheric plasmas generated RONS inactivate bacteriophages: Water-enhanced RONS inactivate the bacteriophages, (**B**) 1 and 2 min treatment enhanced RONS level, inactivated bacteriophages by plasma-activated water (**C**) T4, (**D**) Φ174, (**E**) MS2, and (**F**) storage of plasma-activated water. Reprinted with permission from Guo et al. [111] Copyright 2023 AEM Ltd.

**Figure 11 ijms-24-14106-f011:**
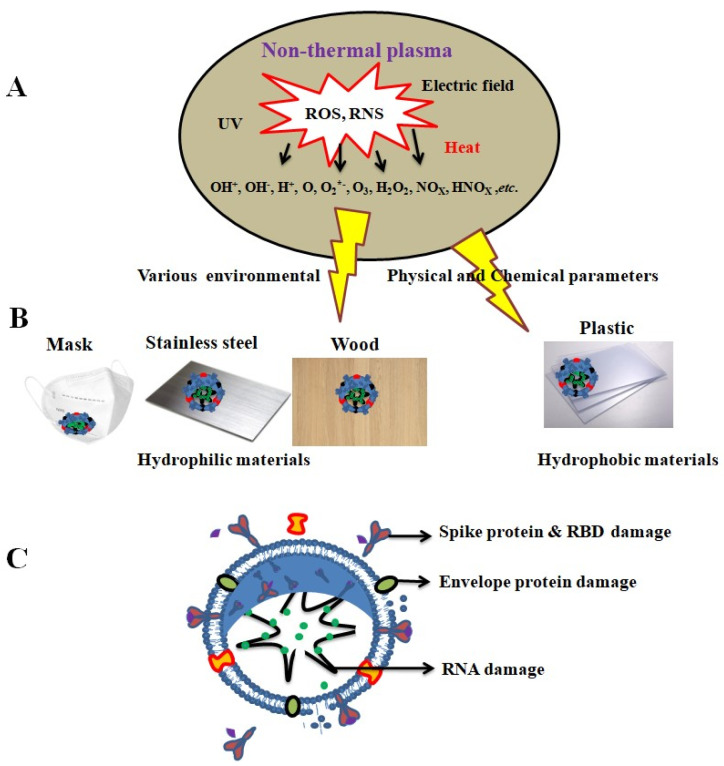
Inactivation of coronaviruses: (**A**) plasma-generated RONS, (**B**) Virus-contaminated materials (Mask, stainless steel, wood, and plastic), (**C**) damaged RBD, spike and enveloped protein, and RNA damage.

**Table 1 ijms-24-14106-t001:** Viral stability on different types of inanimate surface materials with different temperature.

Viral Strain and Cell	Source of Materials	Inoculum Amount	Condition	Period	TCID_50_/mL	Reference
Human Coronavirus 229E, OC43 and embryonic lung cell line L132	Aluminum and Latex surgical gloves	10 µL of a ×10^5^ TCID_50_	21 °C	5 d	90% removed	[51]
Coronavirus 229E and human lung cell MRC-5	Polyfluorotetraethylene (Teflon; PTFE)	×10^3^	21 °C	5 d	ND	[35]
	Polyvinyl chloride (PVC)					
	Ceramic tiles					
	Glass					
	Stainless steel				90% removed	
MERS	Plastic	100 µL of a ×10^6^ TCID_50_	30 °C	48 h	80%	[52]
	Steel					
Human Coronavirus SARS and Vero E6	Wood board	300 µL of a ×10^6^ TCID_50_	Room temperature	60 h	<20%	[53]
	Glass			48 h	51–75%	
	Metal			72 h	26–50%	
	Cloth			72 h	26–50%	
	Filter paper			72 h	26–50%	
	Plastic			60 h	26–50%	
SARS-CoV-1 and Vero E6	Plastic	10^3.4^ TCID_50_	21–23 °C	72 h	10^0.7^	[16]
	Stainless steel	10^3.6^ TCID_50_		48 h	10^0.6^	
SARS-CoV-2 and Vero E6	Plastic	10^3.7^ TCID_50_		72 h	10^0.6^	
	Stainless steel	10^3.7^ TCID_50_		48 h	10^0.6^	
SARS-CoV-2 and Vero E6	Steel	200 µL × 10^6^ PFU	Room temperature	5 ds		[54]
SARS-CoV-2 and Vero cells	Plastic	50 µL of a 1.5 × 10^6^ TCID_50_	Room temperature	168 h	2 log_10_	[55]
	Stainless steel					
	Glass					
	Wood					
	Surgical mask					
	Latex gloves				1.8 log_10_	
	Cotton				1.5 log_10_	
	Paper					
	Ceramics					
Influenza A(H1N1)pdm09 and Madin Darby Canine Kidney (MDCK) cell	Wood	500 µL of 5.5 log_10_	Room temperature	48 h	2.8	[56]
	Plastic			24 h	2.6	
	Stainless steel				2.7	
	Cloth			8 h	1.5	
Influenza Cal/7/09/H1N1 and Madin Darby Canine Kidney cells (MDCKs)	Cotton	10 µL of a 3.3 × 10^4^ pfu/mL	Room temperature	17.7 h	1.2 × 10^3^	[57]
	Microfibre			34.3 h	1.2 × 10^3^	
	Stainless steel			174.9 h	1.9 × 10^3^	

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
