# Peer review of "Persistence of Coronavirus on Surface Materials and Its Control Measures Using Nonthermal Plasma and Other Agents"

_ijms, 2023, doi:10.3390/ijms241814106_

Round 1
Reviewer 1 Report
The authors are reviewing an interesting topic and the data provided of interest.
They organized the considered materials regarding different particularities that are known to favor survival of the virus such as porosity, adsorption, evaporation, isoelectric point, and environmental conditions, such as temperature, pH, and relative humidity.
Since prevention is the best cure, this article is of great interest.
Viruses are inactivated through chemical and physical methods and the review focuses on both:
· Alcohol, chlorine, and peroxide
· Temperature, pH, humidity, ultraviolet irradiation (UV), gamma radiation, X-rays, ozone, and non-thermal plasma generated reactive oxygen and nitrogen species (RONS).
I find the manuscript clear, well organized and containing a lot of useful informations.
The figures are informative and detailed (perhaps they could be a bit bigger.
I have nevertheless a major concern, the authors claim the manuscript being a review, it is in fact an appraisal of non-thermal plasma efficiency and a technical approach to better use it as well a statement to advocate more research on the topic.
The editor may propose to rename the type of article for instance expert view or other.
By the way, apart from some typos and sometimes a bit long sentences. The manuscript is easy to read.
Thank you for giving me the opportunity to review such an interesting manuscript.
Apart from some typos and sometimes a bit long sentences. The manuscript is easy to read.
Author Response
Point-by-point answers to the reviewer’s comments
Reviewer #1
We would like to thank you for providing your constructive and detailed review comments on our manuscript. Manuscript title “Persistence of coronavirus on surface materials and its control measures using nonthermal plasma and other agents” under publications in the International Journal of Molecular Sciences. The recommendations and advice have helped us to significantly enhance the quality of the manuscript.
By the way, apart from some typos and sometimes a bit long sentences. The manuscript is easy to read.
Response: As for your recommendation, I split the long sentences into small ones. Here sentences are in black before splitting and sentences are in red after splitting. (The reviewer’s comments are in black and the author's response to the comments is in red; corrections made by the authors in the main manuscript: are in red).
- Previously: Kasloff et al. [17] reported SARS-CoV-2 survival based on porous and non-porous materials; the N-100 and N-95 face masks reduced the log number 3 in SARS-CoV-2 virus titer within 4 to 7 d, although survival was 14 to 21 d on polyvinyl chloride (PVC) face masks and stainless steel, 7–14 d on gloves, and 1 d on cotton.
Page No: 3; Line No: 109 to 113: Kasloff et al. [17] reported SARS-CoV-2 survival based on porous and non-porous materials. He pointed out, the N-100 and N-95 face masks reduced the log10 number 3 in SARS-CoV-2 virus titer within 4 to 7 d, although survival was 14 to 21 d on polyvinyl chloride (PVC) face masks and stainless steel, 7–14 d on gloves, and 1 d on cotton.
- Previously: The NH2, NH3+, COOH, and COO- groups of amino acids in the SARS-CoV-2 surface-encapsulated S protein (positive charge) mediate solid adsorption through double electrostatic interactions and hydrogen bonding between the ionized surface-active species of the virion and oppositely charged surfaces (hydroxyl-containing surfaces, such as wood, cotton, or paper) (Figure 2 iv).
Page No: 4; Line No: 152 to 156: The NH2, NH3+, COOH, and COO- groups of amino acids in the SARS-CoV-2 surface-encapsulated S protein (positive charge) mediate solid adsorption through double electrostatic interactions and hydrogen bonding. Between the ionized surface-active species of the virion and oppositely charged surfaces (hydroxyl-containing surfaces, such as wood, cotton, or paper) (Figure 2 iv).
- Previously: The face mask helps to control the spread of the virus from respiratory droplets released through sneezes or coughs, and the World Health Organization (WHO) and Centers for Disease Control & (CDC) recommend wearing a mask to prevent SARS-CoV-2 [29].
Page No: 6; Line No: 217 to 219: The face mask helps to control the spread of the virus from respiratory droplets released through sneezes or coughs. The World Health Organization (WHO) and Centers for Disease Control & (CDC) recommend wearing a mask to prevent SARS-CoV-2 [29].
- Previously: They cut the stainless steel into 1 cm2 piece and washed it with 0.01% Tween 80, and 70% ethanol, and cleaned it once with sterile distilled water, and the surfaces were put with 10μl of 10-4 to 10-5 test viruses SARS-CoV in different environmental conditions (temperature 4 °C, 20 °C, and 40 °C and RH 20% ± 3%, 50% ± 3%, and 80% ± 3%).
Page No: 10; Line No: 328 to 332: They cut the stainless steel into 1 cm2 piece washed it with 0.01% Tween 80, and 70% ethanol, and cleaned it once with sterile distilled water. The cleaned surfaces were put with 10μl of 10-4 to 10-5 test viruses SARS-CoV were kept in different environmental conditions (temperature 4 °C, 20 °C, and 40 °C and RH 20% ± 3%, 50% ± 3%, and 80% ± 3%).
- Previously: The England-2 variant SARS-CoV-2 was artificially prepared and sprayed using a 3-jet collision nebulizer into a 40 L Goldberg drum controlled by an AeroMP system (BiAera), the variant attained a decay rate of 0.91%/min in medium RH (40–60%) and high RH 68–88% reached 1.59%/min decay rate, and artificial saliva was more stable at high RH (decay rate of 0.40%/min) than at moderate RH (decay rate of 2.27%/min) [64].
Page No: 10; Line No: 336 to 341: The England-2 variant SARS-CoV-2 was artificially prepared and sprayed using a 3-jet collision nebulizer into a 40 L Goldberg drum controlled by an AeroMP system (BiAera). The variant SARS-CoV-2 attained a decay rate of 0.91%/min in medium RH (40–60%) and high RH 68–88% reached 1.59%/min decay rate, and artificial saliva was more stable at high RH (decay rate of 0.40%/min) than at moderate RH (decay rate of 2.27%/min) [64].
Reviewer 2 Report
Dear editor and authors!
The manuscript summarizes the contemporary state of the art in plasma-based virucidal treatments. The manuscript is relatively well written but it needs further English corrections. However, there are a few flaws and inaccuracies, which have to been corrected before publishing.
Remarks:
Line 67: Do you have Literature for the statement?
Line 111: Literature?
Chapter Adsorption and Evaporation (Line 113 – 136): You describe the measurements of contact angles on smooth and porous surfaces. Do you think these measurements are comparable and the forces behind the spread of a water drop are the same on those surfaces? Please give the reader that information. Keep in mind: on a porous surface a contact angle smaller than 90° does not necessarily describe hydrophilicity.
Line149: Double electrostatic forces? Is it ambiguous? Did you mean double layer forces? Please explain!
Line 151: Mutual attraction have also been observed between likely charged surfaces. Please check the book “Intermolecular and Surface Forces” of Jacob Israelachvili for deeper insights.
Line 248: The citation. You cite the literature under the Name “Katja”. I think it is the first name of the author. Please correct these mistakes (I saw several European first names throughout your manuscript) throughout the whole manuscript.
Line 432 and 436: European first names. I know. We have the same problem vice versa!
You sometimes switch between the log10-representation and the percentage of your reduction due to a virucidal treatment. Please show the results either in the log10 representation or in the percentage of survivors. If not applicable, you´ll show the reader both values. Write log10 (if log10 is used) not just log.
I recommend publishing the paper in the MDPI-Journal “International Journal of Molecular Science” after major corrections.
it needs further English corrections
Author Response
Point-by-point answers to the reviewer’s comments
Reviewer #2
We would like to thank you for providing your constructive and detailed review comments on our manuscript. Manuscript title “Persistence of coronavirus on surface materials and its control measures using nonthermal plasma and other agents” under publications in the International Journal of Molecular Sciences. The recommendations and advice have helped us to significantly enhance the quality of the manuscript.
Response: As per your suggestions all queries have been duly answered. (The reviewer’s comments are in black and the author's response to the comments is in red; corrections made by the authors in the main manuscript: are in red).
1. Line 67:Do you have Literature for the statement?
Response: Page No: 4 & 7; Line No: 69 and 267: Yes, Reference No. 18 and 50 illustrate the persistence of viruses in different materials under various environmental conditions.
2. Line 111:Literature?
Response: Page No: 3; Line NO: 109: Already included Kasloff et al. [17] reference
3. Chapter Adsorption and Evaporation (Line 113 – 136):You describe the measurements of contact angles on smooth and porous surfaces. Do you think these measurements are comparable and the forces behind the spread of a water drop are the same on those surfaces? Please give the reader that information. Keep in mind: on a porous surface a contact angle smaller than 90° does not necessarily describe hydrophilicity.
Response: Page No: 4; Line no 137 to 141: The contact angle and evaporation rate are different between the materials (porous and non-porous surfaces) and different timing intervals. In porous materials droplet was absorbed more quickly compare to non-porous materials, this is depend on ambient environment. Virus persistent are co-related with type of the materials and contact angle. In this article section of adsorption and evaporation section explained in detail.
4. Line149:Double electrostatic forces? Is it ambiguous? Did you mean double layer forces? Please explain!
Response: The intermolecular force (electrostatic forces/interactions) between the material and the virus is a key determinant of the persistence of viruses on materials. Double electrostatic force means, the electrostatic force or interaction between materials and virus. Virus droplets on surface of the materials (liquid + virus), these droplets have an electrostatic force in the material and another electrostatic force on the virus. Actually this is electrostatic interactions (double electrostatic interaction), not in ambiguous and double layer forces.
5. Line 151:Mutual attraction have also been observed between likely charged surfaces. Please check the book “Intermolecular and Surface Forces” of Jacob Israelachvili for deeper insights.
Response: As for your suggestions, I read the Third Edition of Intermolecular and Surface Forces and Editor by Jacob N. Israelachvili.
6. Line 248:The citation. You cite the literature under the Name “Katja”. I think it is the first name of the author. Please correct these mistakes (I saw several European first names throughout your manuscript) throughout the whole manuscript.
Response: Page No: 7; Line No: 254: The author's name changed from Katja, et al [48] to Schilling-Loeffler Katja, et al [48]
7. Line 432 and 436:European first names. I know. We have the same problem vice versa!
Response: Page No: 12; Line No: 439: The author's name changed from Manuela et al. [84] to Buonanno Manuela et al. [84]
Response: Page No: 12; Line No: 443: Author's name changed from Miriam et al. [85] to Darnell Miriam et al. [85]
8. You sometimes switch between the log10-representation and the percentage of your reduction due to a virucidal treatment. Please show the results either in the log10 representation or in the percentage of survivors. If not applicable, you´ll show the reader both values. Write log10(if log10 is used) not just log.
Response: Throughout the manuscript changed from log to log10
Line no: 110, 204, 206, 209, 213, 214, 251, 258, 259, 265, 266, 273, 275, 280, 326, 332, 334, 344, 372, 376, 382 and 425